# Advances in the Immunology of the Host–Parasite Interactions in African Trypanosomosis, including Single-Cell Transcriptomics

**DOI:** 10.3390/pathogens13030188

**Published:** 2024-02-20

**Authors:** Boyoon Choi, Hien Thi Vu, Hai Thi Vu, Magdalena Radwanska, Stefan Magez

**Affiliations:** 1Laboratory for Biomedical Research, Department of Environmental Technology, Food Technology and Molecular Biotechnology KR01, Ghent University Global Campus, Incheon 21985, Republic of Korea; boyoon.choi@ugent.be (B.C.); hien.vuthi@ghent.ac.kr (H.T.V.); vuthi.hai@ghent.ac.kr (H.T.V.); magdalena.radwanska@ghent.ac.kr (M.R.); 2Brussels Center for Immunology (BCIM), Department of Bioengineering Sciences (DBIT), Vrije Universiteit Brussel (VUB), 1050 Brussels, Belgium; 3Department of Biochemistry and Microbiology WE10, Ghent University, 9000 Ghent, Belgium; 4Department of Biomedical Molecular Biology WE14, Ghent University, 9052 Ghent, Belgium

**Keywords:** African trypanosomes, immunity, organ, transcriptomics, single-cell sequencing

## Abstract

Trypanosomes are single-celled extracellular parasites that infect mammals, including humans and livestock, causing global public health concerns and economic losses. These parasites cycle between insect vectors, such as tsetse flies and vertebrate hosts, undergoing morphological, cellular, and biochemical changes. They have remarkable immune evasion mechanisms to escape the host’s innate and adaptive immune responses, such as surface coat antigenic variation and the induction of the loss of specificity and memory of antibody responses, enabling the prolongation of infection. Since trypanosomes circulate through the host body in blood and lymph fluid and invade various organs, understanding the interaction between trypanosomes and tissue niches is essential. Here, we present an up-to-date overview of host–parasite interactions and survival strategies for trypanosomes by introducing and discussing the latest studies investigating the transcriptomics of parasites according to life cycle stages, as well as host cells in various tissues and organs, using single-cell and spatial sequencing applications. In recent years, this information has improved our understanding of trypanosomosis by deciphering the diverse populations of parasites in the developmental process, as well as the highly heterogeneous immune and tissue-resident cells involved in anti-trypanosome responses. Ultimately, the goal of these approaches is to gain an in-depth understanding of parasite biology and host immunity, potentially leading to new vaccination and therapeutic strategies against trypanosomosis.

## 1. Introduction

Trypanosomes are parasites of the order Kinetoplastida, family of the Trypanosomatidae, genus *Trypanosoma*. Trypanosomes are usually transmitted by blood-feeding insects and cause a disease called trypanosomosis in various mammals. Trypanosomes can be classified into two groups. The first group is the “salivarian” trypanosomes. They develop in the anterior part of the insect digestive tract, are present in insects’ saliva, and inoculate the host during feeding. Pathogenic *salivarian* trypanosome species include *Trypanosoma brucei*, including *T. (brucei) evansi*, *T. congolense*, and *T. vivax*. They are extracellular parasites that cause disease in humans (*T. b. gambiense* and *T. b. rhodesiense*), livestock, and game animals. *T. equiperdum* is a sexually transmitted parasite very closely related to *T. (b.) evansi*, and only diverts from other salivarian trypanosomes through its unique transmission mode. The disease caused by all these parasites is often referred to as “African trypanosomosis” (AT). However, animal infections occur beyond Africa and affect regions in Asia, Latin America, and occasionally even Europe [1,2,3]. Human African trypanosomosis (HAT) mainly affects farming communities in rural areas, where its insect vector, the tsetse fly, is found [4]. This disease, also called sleeping sickness, causes fever, headaches, extreme fatigue, aching muscles and joints, and neurologic complications that result in the death of the patient if left untreated [5]. Similarly, AT causes fever, anemia, progressive weakness, weight loss, nervous symptoms, and eventual the death of trypanosusceptible animals such as cattle and horses [3]. It also causes milk and meat production losses, resulting in severe food and economic havoc in affected regions [6].

The second group, the *Stercoraria*, includes *Trypanosoma cruzi*, which develops in the posterior part of the digestive vector tract. This parasite is transmitted to the host when excrements of infected insects come in contact with the host’s damaged skin, oral, or eye mucous membranes. *Trypanosoma cruzi* is an intracellular parasite that causes Chagas disease in animals and humans, mainly in rural areas of Latin America [7]. Due to the intracellular nature of the infection, Chagas’s immunology is completely different from that caused by the extracellular salivarian trypanosomes, and this paper focuses on the latter only.

Unlike other blood-borne protozoa, salivarian trypanosomes remain outside the cell throughout their life cycle. It means that they must overcome continuous exposure to the humoral immune response of their mammalian hosts. When trypanosomes are initially inoculated into a host, a series of events related to innate and adaptive immunity is triggered. Therefore, trypanosomes have evolved to cope with the immune system, allowing them to coexist with their hosts. For example, parasite populations have a remarkable antigenic variation capacity, allowing them to evade host antibody responses for months or years. Additionally, parasite strategies that allow chronic infection are crucial to ensure optimal transmission capacity [8,9]. Therefore, trypanosomosis shows immunopathological phenomena that lead to significant alterations in the immune system and tissues, such as splenomegaly and destruction of lymphoid tissue architecture [10,11,12]. Due to this severe impact on the host immune system, there is still no method to intervene in trypanosomosis infection and transmission through effective anti-parasitic vaccination strategies. Therefore, infection prevention and control must rely on active case diagnosis, treatment, and vector control.

The host’s immune system protects against trypanosome infection through processes such as recognition of pathogens, recruitment of effector cells, and destruction and clearance of the pathogens. Since RNA is essential for determining the functional phenotype of each cell by contributing to the biological pathways in cells and cell–cell communication, studying transcripts is paramount in knowing the host’s immune response to trypanosome infection. Next-generation sequencing (NGS) technologies, including high-throughput RNA sequencing (RNA-seq), such as bulk RNA-seq, enable the study of transcriptional expression at the genome-scale. Moreover, the introduction of single-cell RNA sequencing (scRNA-seq), which can examine the transcriptional expression of individual cells, allows to see a more detailed structural and functional heterogeneity of immune cell and parasite populations [13]. Therefore, sequencing technologies make it possible to discover new gene regulation patterns and cell subtypes and gain knowledge of biological processes that govern trypanosome infections and host–parasite interactions that determine disease outcomes. Accordingly, the most recent studies have addressed host immune responses to AT using high-throughput RNA-seq, even in an organ architectural context (so-called spatial transcriptomics). Therefore, this paper will cover the latest insights into how the host immune system responds to African trypanosomes at the innate and adaptive immune response levels and within various tissues.

## 2. Life Cycle of Trypanosomes

The subgenus of African trypanosomes includes *Trypanozoon*, *Duttonella*, and *Nannomonas*, which account for most pathogenic human and animal infections. In addition, African trypanosomes can be divided into two categories according to their life cycle, i.e., those propagated through cyclical transmission (involving a developmental cycle inside the definitive insect host) and those using rapid mechanical transmission by bloodsucking vectors.

*Trypanosoma brucei brucei* and *Trypanosoma (b.) evansi* are two very closely related parasites and are only differentiated by the fact that the latter has adopted the capacity to bypass the need for an obligatory life cycle step inside the tsetse fly (genus Glossina) [14]. Therefore, as a ‘complete’ paradigm lifecycle of trypanosomes, *T. b. brucei* is usually considered. During a biting event, this cycle starts when an infected tsetse fly injects metacyclic trypomastigotes from its salivary glands into a mammalian host. Once in the mammalian blood, trypanosomes rapidly transform into proliferating ‘long slender’ bloodstream forms [15]. Quorum-sensing signals ensure the arrest of excessive proliferation, by parasite differentiation into nondividing “short stumpy” bloodstream forms, that can only survive if they are able to re-enter the tsetse vector [16]. When the stumpy form trypanosome infects the tsetse, it differentiates into procyclic trypomastigotes in the mid-gut of the fly, proliferates by binary fission, and eventually resists the alkaline enzyme-rich environment of the digestive system. This transition has recently been studied in detail by scRNA-seq [17]. The parasites then transform into long and short epimastigotes through asymmetrical division, of which the short epimastigotes move to the salivary glands and undergo final differentiation into quiescent metacyclic trypomastigotes, ready to infect a new host [18]. ScRNA-seq analysis of this transition has shown a modulation of the parasite metabolism, shifting from a tricarboxylic acid metabolism to the glycolytic metabolism that is ideally adapted to survival in mammalian blood [19,20]. Combined, this mode of propagation is referred to as cyclical transmission. Important to mention is that the metacyclic trypomastigotes undergo meiotic division, a process that now has also been studied at the level of transcriptome changes [20,21], causing increased genetic variability, and making the tsetse the true biological host of the parasite [22,23]. In that sense, the infected mammal could be considered a large parasite reservoir, ensuring that trypanosomes can pass from one fly to the next for prolonged periods of time. Hence, the geographic distribution of *T. brucei* is related to the habitat of the tsetse in sub-Saharan Africa, also referred to as “the tsetse belt”. It is noteworthy that *T. brucei* has three morphologically identical subspecies, i.e., *T. b. brucei*, *T. b. rhodesiense*, and *T. b. gambiense.* While the first one can only infect domestic and game animals due to its inability to cope with the innate trypanocidal factors found in human serum, both *T. b. rhodesiense* and *T. b. gambiense* have evolved into human pathogens, by acquiring various resistance mechanisms that block these innate serum defenses [14,24,25] and will be discussed below.

Recent investigations into the mammalian stage biology of trypanosomosis have shown that particular forms of the parasite can reside in either the skin or adipose tissue, referred to as skin tissue forms (STFs) and adipose tissue forms (ATFs) [26,27]. Interestingly, scRNA-seq has shown that when growth-arrested metacyclic parasites are injected into the skin, transcription is quickly activated, leading to transformation into slow proliferative skin-dwelling trypanosomes that are morphologically similar to bloodstream forms (BSFs) [28]. However, analyzing the gene expression profiles of these parasites indicates that both are not identical [27]. In addition, quiescent slowly replicating motile STFs were found to be capable of repopulating blood, making skin a potential parasite reservoir for long-term infections, as well as a transmission reservoir. Similarly, adipose tissue has recently been proposed to act as a potential *T. brucei* reservoir, harboring replicating, and infective parasites [26]. Interestingly, RNA-seq analysis of ATFs has indicated that these parasites use fatty acid β-oxidation as a process of energy production through fatty acid catabolism. In the context of infection-associated pathology, it should be noted that consumption of fatty acids as a carbon source could contribute to weight loss, a hallmark of AT, and has also been described in experimental *T. brucei* infections in the past [26,29]. In addition, an increase in adipocyte lipolysis through adipose triglyceride lipase activity has been reported to be responsible for the loss of fat and adipose mass caused by *T. brucei*. Here, the immune response to the presence of parasites in tissues was shown to be an important driver of adipocyte lipolysis. Surprisingly, adipocyte lipolysis limited parasite growth and controlled local parasite burden [30] (Figure 1).

*T. (b.) evansi*, which causes the disease ‘Surra’ in mammals such as horses and cattle, is a parasite that is genetically directly derived from *T. b. brucei*. This evolutionary step has occurred many times over, in different locations of sub-Saharan Africa, ensuring that the parasite can bypass the need for tsetse transmission. Indeed, due to the partial or complete loss of kinetoplast maxi-circle and mini-circle DNA *T. (b.) evansi* cannot generate all the gene products necessary to perform oxidative phosphorylation in the midgut of the tsetse [3,31,32]. Hence, this parasite is “trapped” in the long slender bloodstream form but can now be mechanically transmitted through multiple bloodsucking insects, such as biting flies belonging to the families Tabanidae and Chrysops. As a result, *T. (b.) evansi* is found beyond the tsetse belt and is still expanding its worldwide geographic distribution [2,3].

*Trypanosoma equiperdum*, which itself is closely related to *T. (b.) evansi*, is sexually transmitted in the Equidae family, including horses and donkeys, and has, therefore, also reached a wide geographic distribution. Despite the non-salivarian transmission, due to its genetic background, the parasite is still grouped with *T. b. brucei* and *T. (b.) evansi*. The disease caused by *T. equiperdum* is referred to as Dourine [14,24].

The third trypanosome disease of livestock, referred to as Nagana, is mostly caused by either *T. vivax* (*Dutonella*), found in Africa and Latin America or *T. congolense* (*Nannomonas*) that is restricted to the African tsetse belt [14,24]. While both parasites can be transmitted through either cyclical or mechanical transmission, it is clear from the geographic distribution that while the latter is rather successful in the case of *T. vivax*, cyclic tsetse transmission is the primary infection mode for *T. congolense* [33].

## 3. Features of Trypanosomes

As extracellular parasites, trypanosomes are exposed to the mammalian host’s innate immune responses and humoral immune responses. Hence, they should resist all components of the host defense, including antibodies. Therefore, the parasites in the mammalian bloodstream use the variant surface glycoprotein (VSG) as a first line of defense [34]. VSGs form a monolayered, thick, dense surface coating, covering the entire plasma membrane, anchored by glycosylphosphatidylinositol (GPI) anchors. Using the GPI anchoring mode, VSGs are able to scavenge host antibodies, and efficiently ‘sort’ and endocytose VSG-antibody immune complexes through the flagellar pocket. This prevents antibodies from executing their primary function, i.e., clearing infection and complement-mediated attack of the parasite [35,36,37]. However, since the host antibodies easily recognize trypanosome surface antigens, the parasite co-evolved an elaborate mechanism of antigenic variation using an extended battery of genes encoding for the VSGs. Since African trypanosomes can only use one *VSG* variant at a time by so-called mono-allelic expression, these surface proteins avoid antibody-mediated parasite clearance by continuously changing. Using an scRNA-seq approach, *VSG* expression was shown to start with metacyclic parasites individually expressing transcripts, encoding for a unique metacyclic VSG coat glycoprotein [20]. Next, the expression of *VSG* genes in the BSFs is carried out through the use of a so-called expression site (ES), a telomeric region that contains beside the VSG-encoding region, a number of expression site-associated genes (ESAGs), that also play a role in the interaction with the host. Several known mechanisms drive antigenic variation in trypanosomes. The first mechanism is transcriptional switching, which turns off the elongation of transcription of the active ES and turns on the silent ES. The second mechanism is homologous recombination, in which the active *VSG* gene can be changed. Homologous recombination includes gene conversion by changing the active *VSG* gene and another *VSG* gene from a repertoire and reciprocal exchange by replacing a copy of the active *VSG* gene with a new copy derived from another ES [15]. A third mechanism that relies on the genetic rearrangement of the trypanosome genome is segmental gene conversion. This can create novel antigens, mosaic VSGs, by recombining pseudogenes or various silent *VSGs*. These mosaic VSGs allow the creation of diverse VSG antigens beyond the existing *VSG* repertoire. The existence of such an inexhaustible VSG generation mechanism has been studied in detail at the genetic level [9,38,39]. In conclusion, creating countless *VSG* variants through these mechanisms will make antigen recognition by the host’s adaptive immune system difficult and enable immune evasion by the trypanosome.

What is important to comprehend is that if trypanosomes were able to evade host immune responses completely, this would rapidly lead to mammalian death, hence eliminating the feeding reservoir for the tsetse. This would not benefit the parasite. Therefore, trypanosomes have adopted a self-regulated growth control system, called quorum-sensing. This mechanism drives trypanosomes from actively dividing slender forms to non-dividing stumpy forms [40]. The basis of population density sensing is related to oligopeptides, derived from hydrolyzed host or trypanosome proteins generated by the activity of peptidases released by the slender forms. Subsequently, these peptides are sensed by the surface-expressed oligopeptide transporter GPR89, leading to trypanosome growth arrest [41]. In addition, quorum sensing may be involved in controlling the peak of parasitemia by monitoring the inflammatory state of the host [16,42]. Recently, the process of quorum sensing in *T. brucei* was assessed through scRNA-seq, comparing slender, intermediate, and stumpy bloodstream-form parasites [43]. Here, trajectory analysis allowed to identify dynamic gene expression patterns during differentiation, indicating that slender forms that have completed the cell cycle, exit the latter, subsequently enter the G0 phase, and finally differentiate into stumpy forms, with ZC3H20 being an essential regulator in this process [43].

## 4. Human African Trypanosomosis

*T. b. rhodesiense* and *T. b. gambiense* both cause sleeping sickness. Due to acute and fatal virulence, *T. b. rhodesiense* HAT accounts for only 2% of all HAT cases and is restricted to East Africa. *T. b. rhodesiense* is considered a zoonotic parasite, with a persistent animal reservoir. Therefore, it is unlikely that *T. b. rhodesiense* HAT will be eradicated soon. *T. b. gambiense* HAT on the other hand causes chronic infection and has historically accounted for 98% of all HAT cases, occurring mostly in West and Central Africa. *T. b. gambiense* is considered anthroponotic, meaning that human-to-human transmission is the main route of disease spread [24]. Due to recent successes in gambiense HAT surveillance and treatment, this form of sleeping sickness will probably no longer be considered a human health threat in sub-Saharan Africa by 2030 [44]. HAT infections can be divided into two stages. The first stage is the hemolymphatic stage, where the parasite invades the host’s circulatory and lymphatic systems and causes immune dysfunction [5]. The second is the meningoencephalitis stage, in which parasites invade the blood–brain barrier and cause neuropsychiatric manifestations, including daytime somnolence and nocturnal insomnia. Ultimately, this will lead to the comatose death of the patient if the disease is left untreated [5,45].

Humans are mostly resistant to *T. b. brucei* as well as to *T. (b.) evansi*, *T. equiperdum*, *T. vivax*, and *T. congolense*, due to the presence of innate trypanolytic components in serum. This lytic activity is also observed in the serum of some other primates, such as *G. gorilla* (Western lowland gorilla), *P. papio* (Guinea baboon), *P. cynocephalus* (Yellow baboon), and *P. anubis* (Olive baboon) [46]. In human serum, innate trypanosome resistance is driven by high-density lipoprotein (HDL) particles, containing the trypanolytic factor 1 (TLF1) and TLF2. These particles contain haptoglobin-related protein (HRP) and apolipoprotein L1 (APOL1), and in the case of TLF2 also immunoglobulin (Ig)Ms. At least for TLF1, the entire mode of action in killing *T. b. brucei* has been fully elucidated. After HRP binds to hemoglobin, the TLF complex is captured by a trypanosome receptor in the flagellar pocket of the trypanosome. Subsequently, it is internalized into trypanosome digestive organelles, where degraded hemoglobin and other host components present in the liquid phase can be used for new biosynthetic processes [15,47]. In *T. brucei*, the specific receptor Haptoglobin-hemoglobin receptor (HpHbR) mediates the uptake of TLF1 [48]. The uptake of TLF2 is associated with IgM scavenging, albeit the exact process has yet to be elucidated [49,50]. Following binding, the particles pass through the endocytic pathway of early and late endosomes (neutral and acidic pH, respectively) and lysosomes (acidic). The acidic pH of lysosomes changes the shape of the membrane-addressing domain, one of the three domains of APOL1, causing it to dissociate from HDL particles and bind to the lysosomal membrane. Afterward, APOL1 forms ionic pores in the lysosomal membrane, and as a result, osmotic swelling occurs, and intracellular pressure increases, ultimately causing cell death [15]. The resistance of *T. b. rhodesiense* to the lytic action of APOL1 is related to the presence of the serum resistance-associated (SRA) protein, encoded by an ESAG, which can block the pore-forming conformational change of APOL1 [15]. The resistance mechanics of *T. b. gambiense* is more complex and divides trypanosomes into group 1 and group 2 parasites. Group 1 *T. b. gambiense* shows consistent human serum resistance and slight genetic variation and has the *T. b. gambiense*-specific glycoprotein (TgsGP) as a genetic marker [51]. Group 2 *T. b. gambiense* is more heterogeneous and does not have a specific marker [52]. Neither group 1 nor group 2 have the *T. b. rhodesiense SRA* gene [53]. However, group 1 parasites resist the lytic activity of human serum by exhibiting a specific point mutation in HpHbR, reducing the binding and uptake rate of TLF1 [48,54]. It has also been proposed that TgsGP stiffens the trypanosome lysosomal membrane and thus prevents the insertion of APOL1. These parasites also have increased cysteine protease activity, which reduces the trypanolytic action of APOL1 [55]. Finally, the mechanism for human serum resistance in group 2 remains elusive at this moment and requires further investigation.

Recently, RNA-seq transcriptomic approaches have been used to provide new insights into immunobiology responses in the blood and central nervous system of *T. b. rhodesiense* HAT patients [56]. Here, differentially expressed genes were identified in so-called “early stage” blood versus healthy controls, as well as ‘early stage’ blood versus ‘late-stage’ cerebrospinal fluid (CSF). This comparison identified mainly innate immune responses during the early stage of infection, based on up-regulation of the classical complement pathway genes (*C1QA*, *C3Ar1*, *CR1*) and pro-inflammatory tumor necrosis factor (TNF)-α-induced proteins (*TNFAIP6*, *TNFAIP8*). In addition, *C1QC*, encoding a part of the 1q complement component, *MARCO*, a scavenger receptor of macrophages for cleaning pathogens, and *IGHD3-10*, encoding the immunoglobulin heavy chain diversity antigen receptors, were upregulated more than 5-fold in the blood of early stage patients [57,58,59]. In contrast, in the last stage of infection, anti-inflammation and neuro-degeneration pathways were activated through enrichment of *interleukin (IL)-10*, *ADCY2*, and *AKAP*, of which the latter has also been related to bipolar disorder and schizophrenia [60,61], and *GABRA2*, *GABRB*, and *ApoB* that have been associated with sleep disorders [62,63].

## 5. Host Immune Responses to Trypanosomes

The mammalian host’s innate and adaptive immune systems are both key to successfully resisting or controlling trypanosomosis. When trypanosomes are inoculated into the mammalian hosts by a blood-feeding insect such as a tsetse fly, the first contact between the trypanosome and host occurs in the skin. Here, a chancre often develops at the dermal inoculation site. Intense innate immune reactions, cellular reactions, and edema formation accompany these chancres [64,65,66,67,68]. Thereafter, parasites start to circulate through the blood or lymph, invading lymphatic tissues and various organs [69,70,71]. There, the trypanosomes again encounter various innate immune components before being confronted with the adaptive immune system. Once entered into the circulation stage of infection, trypanosomes are going to encounter responses from macrophages and B cells, as well as the T helper compartment that links these two [69,72].

### 5.1. Macrophages and Their Cytokines

The secreted types and roles of cytokines in the pathogenesis of AT vary depending on the production quantity and the phase of infection. AT affects the host in different ways, with susceptibility and resistance to the disease determined by the host’s cytokine balance during infection. Various studies have shown that African trypanosomes induce many cytokines in mammalian hosts. Dominant cytokine producers are macrophages, the primary cells contributing to innate immunity that phagocytose pathogens [73]. Macrophages are innate antigen-presenting cells that are essential in T-cell activation [74] and, hence, are indirectly involved in adaptive immune responses. This transition function is mediated through the secretion of effector molecules, including cytokines [75], that are also involved in immunosuppression and immunopathology.

Activated macrophages can be largely classified into (i) classically activated macrophages that secrete proinflammatory cytokines such as IL-1, IL-6, and TNF, and produce nitric oxide synthase, an enzyme necessary to deliver nitric oxide, or (ii) alternatively activated macrophages, that produce anti-inflammatory cytokines such as IL-4, IL-10, and IL-13 [75]. Classically activated macrophages are typically recruited to fight pathogens by promoting inflammation in early infections. To counterbalance this activity, alternatively activated macrophage induction is a hallmark of chronic infections, as these cells have an immunosuppressive function, reduce inflammation, and prevent and repair tissue damage [75].

During trypanosome infections, macrophages are activated by a number of parasite compounds, including the VSG-GPI anchor and unmethylated trypanosomal DNA, released from dead trypanosomes. Activation is enhanced by interferon-gamma (IFN-γ) produced by T-helper (Th) 1 cells. Activated macrophages play a significant role in controlling the parasitemia peak by secreting proinflammatory cytokines such as TNF, IL-12, and nitric oxide [76,77], as well as parasite phagocytosis [72,78,79]. Interestingly, TNF is not only involved in the inflammatory response in the early phase of trypanosomosis but is also involved in the induction of chronic anemia and the neurologic signs in the late phase of infection [80]. The role of TNF in the inflammation of the central nervous system lesions, found in chronic *T. b. brucei*-infected mice, has been known for a long time [81]. It is, however, only much more recently that scRNA-seq analysis has provided a detailed view of overall mechanisms that drive brain pathology in experimental models for AT. This new data has indicated a possible role of microglia cells, the primary immune cells of the central nervous system that are similar to peripheral macrophages, in driving pathology [82]. These new findings indicate changes in innate and adaptive immunity, the type I interferon response, neurotransmission, synaptic plasticity, pleiotropic signaling, circadian activity, and vascular permeability, all correlating with early central nervous system (CNS) symptoms during the encephalitic stage [83]. Furthermore, immune alterations through interactions between microglia and other immune cells were discovered, exemplified by the crosstalk between microglia and CD138^+^ plasma cells. Here, homeostatic microglia actively support CD138^+^ plasma cell survival via B cell activating factor (BAFF), and reciprocally, CD138^+^ plasma cells produce IL-10 to mitigate inflammatory responses in microglia during *T. brucei* infection [82].

Alternatively activated macrophages secrete anti-inflammatory cytokines IL-4, IL-13, and thymic stromal lymphopoietin (TSLP), a key cytokine that promotes Th2 response [84,85]. In mice, IL-4 was required or prolonged survival of *T. b. gambiense* infected mice [86]. Likewise, TSLPR^−/−^ mice were susceptible to *T. congolense* infection by failing to control parasitemia compared to wild-type control. The latter is attributed to impaired activation of alternatively activated macrophages and overproduction of proinflammatory cytokines, including IFN-γ and TNF [85]. Meanwhile, one of the key anti-inflammatory cytokines that regulate African trypanosome infection is IL-10. It reduces the effector activities of T cells and macrophages which produce inflammatory cytokines after initiating a trypanosome infection [87]. In the absence of IL-10, *T. b. brucei*-infected mice maintain an inflammatory response and succumb early [88].

### 5.2. T Cells

Like macrophages, T cells also control AT by secreting cytokines. CD4^+^ T cells regulate inflammatory responses by producing cytokines and assisting B cells with effective isotype class-switching and specific antibody responses to parasite antigens [89]. It has been shown that during *T. congolense* infections, IgG2a, IFN-γ, and IL-10 production were all impaired in CD4^+^ T cell-deficient mice [90]. In addition, CD8^+^ T cells can kill pathogens by secreting cytokines, even in the case of extracellular pathogens such as salivarian trypanosomes. Trypanosome lymphocyte-triggering factor (TLTF), a trypanosome component located in the flagellar pocket, is known to modulate the cytokine network of the host immune system by activating CD8^+^ T cells and inducing the production of IFN-γ [91]. In the early infection stage, the TLTF level increases for interaction with the host. However, continuous release of this molecule over-stimulates the host immune system. This is ultimately unfavorable for the parasite, so the level of TLTF decreases as the host produces anti-TLTF antibodies, blocking the effect of this trypanosome molecule through neutralization in the late stage of infection [92].

In mice infected with *T. b. brucei*, natural killer (NK) and NK-T cells first produced IFN-γ, followed by CD8^+^ and subsequently CD4^+^ T cells. IFN-γ not only participates in the early inflammatory reaction but also activates phagocytosis of red blood cells by recruiting erythrophagocytic myeloid cells, causing anemia, the main hallmark of AT. Hence, the absence of NK, NK-T, and CD8^+^ T cells showed a reduced anemic phenotype during trypanosomosis through cell depletion and neutralization experiments. This observation was confirmed using IFN-γR^−/−^ mice [93]. When progressing into the later stages of infection, CD4^+^ T cells become the main produced IFN-γ [94]. Interestingly, however, CD4^+^ T cells are also crucial producers of IL-10 at this stage, needed for the anti-inflammatory cytokine context alongside *T. brucei*-induced alternatively activated macrophages [95]. As it will be outlined in detail below, the most recent scRNA-seq data has provided new insights into the cellular source of both IFN-γ and IL-10, showing that a unique population of CD4^+^ T cells is capable of producing both ‘counteracting’ cytokines simultaneously [96].

### 5.3. B Cells

As African trypanosomes are extracellular bloodstream parasites continuously exposed to the host’s humoral immune response, they are expected to be targeted by antibodies, and hence activating B lymphocytes. Interestingly, DNA from *T. b. brucei* can induce B cell proliferation [76]. On the B cell side itself, the lymphocyte adapter protein Bam32, important for B cell activation and cell survival, is crucial during trypanosomosis. Indeed, in Bam32^−/−^ mice, the germinal center response associated with the anti-trypanosome IgG antibody response was abrogated, resulting in a worsened control of *T. congolense* infection [97].

As mentioned in the introduction section, trypanosomes use a system of antigenic variation to construct their glycoprotein coat, using a single VSG at any given time [98]. Since trypanosomes can produce antigenically distinguishable VSGs, they force the host to trail the VSG switching with the switching of antibody responses. When specific antibodies against the specific epitopes of the VSG are generated, opsonized parasites, VSG-coated trypanosomes, are mainly phagocytosed by liver resident macrophages, i.e., Kupffer cells [78], thereby clearing successive parasitemia waves [99,100]. In theory, the removal of trypanosomes should be accomplished by a combination of antibody-dependent phagocytosis and complement-mediated lysis, guaranteed by the presence of anti-VSG antibodies. Initially, immunoglobulin levels, especially IgM, increase dramatically in mammalian hosts infected with trypanosomes. When IgMs against the specific VSG are produced, they should be able to fix complement and assist in the assembly of the various complement components on the parasite’s surface, leading to the destruction of the latter [101]. However, a wealth of data has been generated over the years, indicating that trypanosomes can efficiently block the complement cascade from reaching the final membrane attack complex [102].

Activated B cells initiate germinal center formation with the help of follicular CD4^+^ T helper cells and produce various antibody isotypes through class-switching recombination, including IgGs that generally have an increased antigen binding affinity compared to the primary IgMs [89,103]. Hence, IgGs should play an important role in the phagocytosis of trypanosomes by macrophages [104]. However, the exact role of various antibody classes is not clear-cut. Indeed, in a study using a *T. b. brucei* model, IgG was shown to have a dominant role over IgM, when it comes to parasite clearance [105]. In contrast, IgMs were found to play a dominant role when studied in a model system of experimental *T. (b.) evansi* infections in mice [106,107]. Important to mention, however, is that trypanosomes have adopted an efficient mechanism to remove antibodies from their surface, ensuring that immune-mediated elimination never reaches a level of complete parasitemia clearance prior to the switching of the coat by the mechanism of antigenic variation [36]. Hence, while highly specific anti-VSG antibodies can eliminate trypanosomes in a VSG-dependent manner, a small population of trypanosomes that manages to escape an anti-VSG host attack will perpetuate the infection. In order to try and counteract immune escape, the host is triggered into excessively activating antibody-producing B cells, resulting in hypergammaglobulinemia [108] and polyclonal B cell activation [109] This increases the level of polyspecific and non-specific antibodies, including autoantibodies in the serum, in addition to trypanosome-specific antibodies [110]. Trypanosome DNA and VSG appear to contribute to host non-specific B cell activation [111]. Simultaneously, specific antibody response to trypanosome antigens decreases. Therefore, while in the early stages of infection, trypanosome-specific antibodies are mainly involved in parasite clearance, as the infection progresses, a significant portion of antibodies will become polyspecific, and the fraction of autoantibodies against nucleic acids and even red blood cell surface proteins increases [112,113]. Recently, it was confirmed that B cells from *T. brucei*-infected murine meninges produce high-affinity autoantibodies that can recognize mouse brain antigens, including myelin basic protein, which are associated with cortical demyelination and brain pathology [114,115]. This study will be described in detail in the section below (Section 6.5) [116].

During the late stages of trypanosome infection, the B cell compartment becomes suppressed or even depleted. This process involves the rapid disappearance of immature B cells in the bone marrow and transitional and marginal zone B cells in the spleen, followed by the gradual disappearance of follicular B cells in the spleen [12,69,117]. These damaged and missing B cells cannot effectively produce antibodies against new VSG variants. Therefore, IgM and IgG responses are significantly reduced or eliminated [105]. Additionally, a decrease in anti-VSG recall responses to previously encountered VSG variants may allow the re-emergence of previously encountered trypanosome variants and allow the accumulation of mosaic VSG variants [39]. In addition, infection-associated B cell depletion will ultimately reduce the efficacy of vaccines unrelated to trypanosomosis [12,118]. While in the past, events related to the infection-associated abrogation of B cell functionality have been mainly described in a phenotypic and functional cellular context, most recent data has now come from scRNA-seq approaches [107,116,119]. The obtained data will be discussed below in an organ-relevant context.

## 6. Host Immune Responses by Organ

Through multiple investigations over several decades, it has become clear that the pathology of AT is highly complex. For example, the mechanisms for pathogenesis and pathology of AT are extensive and vary depending on the trypanosome strain, host tissue, and stage of infection. While evading the host’s immune system, trypanosomes invade tissues in many organs, such as blood and lymph, causing multiple symptoms. For example, they can cause muscle atrophy, splenomegaly, and later meningoencephalitis in the brain in the case of HAT. Therefore, it is important to discover how African trypanosomes trigger, alter, and evade immune responses in each host organ and, ultimately, by what mechanism the parasite causes disease. In the past, host immune responses to African trypanosome infection were usually observed through antibody assays or protein measurements. However, these methods can only observe stable and abundant proteins and have limitations in detecting transiently expressed or low-abundance proteins. In this context, a transcriptome approach using RNA-seq or scRNA-seq enables more sensitive measurement of RNA transcripts at the genome level. It can lead to a better understanding of proteins and related pathways triggered by immune responses to trypanosome infection. Therefore, the sections below provide up-to-date information on how the host’s innate and adaptive immune systems respond to AT in individual immune organs based on the most recent transcriptomic data available (Figure 2).

### 6.1. Skin

AT in mammals is initiated by a skin bite of an infected insect, often causing a local inflammation referred to as a chancre [64,65,66,67]. Hence, host–parasite interactions at the level of the dermis are crucial during the onset of parasitemia. Immune cells found in the dermis include Langerhans cells, dermal macrophages, dermal dendritic cells, NK cells, and T cells, while the subcutaneous layer mainly comprises adipocytes [120]. Trypanosomes need to be able to pass the dermis in order to migrate toward local lymphatic and blood vessels and differentiate into proliferative trypanosomes in the bloodstream, but a subset of parasites appears to remain at the inoculation site as skin-dwelling trypanosomes [64,121]. Interestingly, both skin and blood parasites contribute to infection and transmission [64,65]. This might be particularly important in the case of human *T. b. gambiense* HAT, that can often be found as latent infections where virtually no blood parasites can be detected, but where skin-dwelling trypanosomes could still constitute a transmission reservoir [122,123]. At the level of the skin, the dermal neutrophil compartment is considered a crucial defense barrier against the invasion of pathogens. In case of trypanosome transmission by an infected tsetse, these cells are rapidly recruited to the bite site [68]. Based on conventional immunology concepts, neutrophils could be important for parasite control by inducing hostile inflammatory conditions or releasing anti-microbicidal factors and neutrophil extracellular traps (NETs) [124]. However, phagocytosis of trypanosomes by neutrophils was rarely observed in the skin, and neutrophil depletion caused a decrease in parasitemia levels in peripheral blood [68]. This observation has more recently prompted a single-cell and spatial transcriptomic investigation to determine changes in skin cell populations and immune–stromal crosstalk during chronic infection with *T. brucei*. Here, the expansion of interstitial preadipocytes (stem-like adipocyte precursor cells) present in subcutaneous adipose tissue and Langerhans cells adjacent to subcutaneous adipose tissue was observed, with these cells participating in local cytokine production and antigen presentation. In addition, special attention was given to the assessment of γδ T cells during infection, as these cells are considered part of the resident dermal immune compartment and are known to be important regulators of tissue homeostasis and tissue repair, expressing a variety of effector molecules, including IL-17A [125,126,127,128]. Interestingly, scRNA-seq analysis and single-molecule fluorescence in situ hybridization (smFISH) validation showed the expansion of dermal IL-17A-producing Vγ6^+^ cells in subcutaneous adipose tissue in a γδ T cell subset [67]. Additionally, potential interactions between subcutaneous adipocytes and Vγ6^+^ cells, involving *Tnfsf18* signaling through a network of adipose-derived ligand *Tnfsf18* and T-cell-specific receptor *Tnfrsf18*, were predicted by cell–cell communication analyses, with the upregulation of *Tnfsf18* in the subcutaneous adipose tissue being confirmed by using smFISH. This interplay has been shown to mediate T cell activation via *Tnfsf18* and promote adipocyte metabolism via Vγ6^+^ cell-derived *Clcf1* and *Areg*. Finally, these in silico predictions were validated based on an increase in activated IFN-γ-producing CD8^+^ T cells and the reduced loss of subcutaneous adipose tissue in Vγ4/6 γδ T cell-deficient mice, compared to wild-type control mice. Combined, these results suggest a role for subcutaneous adipocytes and γδ T cells as homeostatic regulators in the skin during chronic infection with trypanosomes [67].

### 6.2. Spleen

In the case of blood-borne infections, the spleen is a crucial organ, as it executes functions such as the storage and release of immune cells and is important for the production of IgM antibodies by marginal zone B cells (MZBs). The spleen is also involved in removing aging red blood cells, morphologically abnormal red blood cells such as sickled cells, and cells affected by intraerythrocytic parasites such as malaria. Additionally, red blood cells may be produced in the spleen when hematopoiesis fails in the bone marrow due to hematological diseases [129]. When immune reactions are activated, an enlargement of the spleen is often observed, referred to as splenomegaly, affecting the normal function of the organ. This can drive the excessive removal of blood cells from circulation, causing anemia and platelet reduction [129]. Enlargement of the spleen has been observed in patients with *T. b. rhodesiense* [130] and *T. b. gambiense* HAT [131]. The same is observed under experimental conditions in *T. b. brucei*, *T. (b.) evansi*, *T. congolense* and *T. vivax*-infected mice [10,11,132,133]. In mice, splenomegaly becomes evident after the first week of infection, reaching excessive proportions towards the 3rd week, where a 20-fold increase in spleen volume has been reported. Hypersplenism coincides with a significant reduction in the red blood cell count, hemoglobin, and hematocrit values. In addition, when histopathological examination of spleens from infected mice at various stages of trypanosome infections was performed, it was observed that the clear separation of white and red pulp that is present in uninfected tissues collapsed due to infection [11]. These combined data show that African trypanosome infection triggers splenomegaly accompanied by drastic changes in spleen cell numbers. This coincides with a complete abrogation of B cell development, leading to a gradual degradation of spleen B cell numbers [12]. For MZBs, this is already observed at 6 days post infection of *T. brucei*, when the first parasite peak occurs, and when numbers of follicular B cells (FoBs), CD4^+^ T cells, and CD8^+^ T cells temporarily increase, before they also collapse [69]. MZBs, which reside in the marginal zone of the spleen, play an important role in the early stages of antibody response against T-independent antigens and the VSG coat on trypanosome cells and develops into short-lived plasma cells, rapidly producing relatively low-affinity antibodies and mediating phagocytosis for parasite clearance. Therefore, this depletion of the MZBs enables the evasion of the host’s first line of immune response.

Recently, the collapse of the spleen immune compartment has been investigated at the level of the transcriptome using scRNA-seq. As already outlined above (see the section on B cells), these results have shown that *T. (b.) evansi* infection results in the rapid activation of MZBs and FoBs, along with lgG2c expression in FoBs during the early stages of infection. Subsequently, transitional B cell and MZB numbers decreased rapidly, while FoB numbers decreased gradually after control of the first parasitemia peak, and germinal center B cells and plasma cells transiently increased at this point. Finally, the entire B cell population was depleted during infection at late chronic time points. Interestingly, state-of-the-art scRNA-seq analysis has overcome the limitations of conventional flow cytometry visualization, allowing to investigate the effects of infection on minor cell populations such as the transitional B1 B cells, and atypical short-lived memory B cells (atMBCs) [107]. ScRNA-seq analysis has also helped to guide and re-address investigations into the role of IgM and IgG antibodies in trypanosome control, which was previously studied in both *T. b. brucei* [105] and *T. (b.) evansi* models [106], confirming the crucial role of IgMs in the latter [107]. Interestingly, this new approach also revealed a new, unexpected trypanosome defense strategy against the host immune system, which is the rapid induction of a T cell-independent IgG2c antibody class switch to diminish the danger posed by host IgM antibodies. This spleen scRNA-seq-driven hypothesis was subsequently confirmed by using an activation-induced cytidine deaminase-deficient (AID^−/−^) mouse model. Here, it was confirmed that mice that do not undergo an antibody class-switch during infection exhibit superior parasitemia control and maintenance of the splenic B cell compartment during a *T. (b.) evansi* infection.

In addition to its role in B cell activation and maturation, the spleen is also a crucial immune organ for mature CD4^+^ T cell activation and differentiation into various subsets of helper cells. As already indicated above, both CD4^+^ Th1 and Th2 cells are known to play distinct roles during trypanosome infections. Hence, having recent access to spleen scRNA-seq data has allowed to obtain a more detailed view of the transcriptome profile alterations that take place during infection [96]. Comparative data analysis of naïve C57BL/6 splenocytes and cells derived from 14- and 42 days post infection allowed to characterize six CD4^+^ T cell subpopulations affected by experimental *T. (b.) evansi* trypanosomosis, including naïve CD4^+^ T cells, Th1 cells, regulatory T cells (Treg), T follicular helper cells (Tfh), cycling T cells, and newly discovered myeloid cell phenotype expressing T cells. Contrary to expectation, no cells were found with a CD4^+^ Th2 transcriptome profile, not even during the late stage of infection, which is hallmarked by increased plasma concentrations of IL-10. A second surprising finding was that while Tregs were present as expected from earlier *T. congolense* studies [134,135], they did not have an IL-10 regulatory signature, and the proportion of these cells in the total CD4^+^ T cell pool drastically decreased during the chronic stage of infection. Finally, a third surprising result from the scRNA-seq approach was the discovery of a small population of CD4^+^ Th1 cells that simultaneously produced IFN-γ and IL-10. This suggests that the T cell biology of trypanosomosis should be re-addressed with a focus on Th1 cells, the plastic immune regulatory role of these cells, and their contribution to the regulation of survival of the host. Indeed, from this recent data, it appears that these cells are crucial during the early stage of infection by aiding macrophage activation and parasite clearance, while later on in infection, they are partially involved in alleviating cellular activation and ultimately reducing inflammation-induced pathology during infection [96].

ScRNA-seq analysis also allows the deciphering of cell–cell interactions. Hence, this aspect of the immunobiology of trypanosomosis has recently been addressed at the level of the spleen. Results indicate that the Tfh cells that generally play an important role in the activation process of follicular and germinal center B cells through interaction with B cells expressed high levels of *Ctla4*, leading to an inefficient germinal center response through interaction with the CD86 molecule on B cells. In turn, this leads to the abrogation of germinal center formation. Hence, this finding suggests that trypanosomes have developed an additional mechanism to evade antibody-mediated immune attacks by depriving the host of the capacity to generate high-affinity anti-parasite antibodies [96].

A most recent aspect of spleen immunobiology of trypanosomosis has been addressed by scRNA-seq, relates to the previously described neutrophil accumulation observed after the clearance of the first peak of parasitemia [69]. This finding was recently confirmed, showing a 15-fold increase in spleen neutrophil cell numbers by the 2nd week of infection. Hence, this infection time point was used to perform a detailed study of cellular neutrophil heterogeneity at the transcriptome level [119]. In line with the known neutrophil maturation process [136,137,138], four subpopulations (N1–N4) were also found in the *T. b. brucei* infection model. The populations represent pre-neutrophils, immature neutrophils, continued immature neutrophils, and mature neutrophils, respectively. As expected, most neutrophils in non-infected control mice were mature cells, whereas pre-neutrophils and immature cells were observed aside from mature cells in infected mice. Interestingly, no clear infection-induced neutrophil number alteration was observed at the bone marrow level. This suggests that neutrophil differentiation, which usually occurs in bone marrow, is triggered directly in the spleen during *T. b. brucei* infections [119]. This is interesting as neutrophils are considered to be cells that can play a double-edged sword role during infection, helping either in pathogen elimination or being responsible for tissue damage and organ pathology. Both can be associated with their role in innate immune defenses by releasing cytotoxic enzymes from granules, including metalloproteinases (MMP)-8 and MMP-9. These MMPs are able to digest collagen and elastin [139] and promote the movement of neutrophils to the site of inflammation through the extracellular matrix (ECM) [140]. Tissue inhibitors of metalloproteinases (TIMPs) should normally suppress excessive MMP activity under homeostatic conditions [141]. Hence, the recent results of both scRNA-seq analysis and actual protease concentration confirmation assay both have now shown that trypanosomes trigger excessive MMP-8 and MMP-9 release and activation, in particular, involving the N3 subpopulation, in the absence of TIMP inhibitory activity. This ultimately leads to the destruction of the ECM and damage to the splenic follicular architecture and once again aids the parasite by depriving the host of the capacity to generate an immune environment needed for producing high-affinity antibodies [119]. Indeed, trypanosomosis-associated B cell depletion coincides with the neutrophil-driven destruction of spleen germinal center architecture. In this context, and as mentioned above, earlier work had already shown that blocking neutrophil accumulation during trypanosome infections ameliorates pathology and promotes parasitemia control [68]. Hence, in view of the new data outlined here, it could be that the collateral damage caused by activated neutrophils to the host B cells compartment actually helps the parasite, overruling the potential immunoprotective functions of these cells during chronic infection.

As the spleen is a multifunctional immunological organ, macrophage activation is one of the key factors that will aid in the infection control of blood-borne pathogens. For years, experimental trypanosome infections have been used to study these cells, leading early on to the idea of the so-called M1/M2 classification. Here, cells that produce proinflammatory cytokines and contribute to controlling parasitemia were designated as “M1–classically activated”. As these cells can cause infection-induced pathological features such as systemic immune response syndrome, a switch to M2 macrophages that produce anti-inflammatory cytokines is required, leading to the paradigm that the M1/M2 macrophage balance would be important for prolonged survival [75,87,142]. However, recent studies have shown that this cellular classification is insufficient to explain heterogeneous macrophages. Indeed, the transcriptome profiling of macrophages of *T. (b.) evansi*-infected mice, analyzed on day 14 (acute phase) and day 42 (chronic phase), subdivides the spleen macrophage population in (i) monocytes-derived-macrophages (MDM), (ii) erythroblast island macrophages (EBIM), and (iii) tissue-resident macrophages (TRM). However, no discrete M1 and M2 macrophage populations could be distinguished. Instead, genes encoding previously designated as M1 and M2 markers were simultaneously expressed in TRM [132].

Since the spleen is the hematopoietic organ under tissue hypoxia, at least in mice and cattle, anemia is a hallmark of trypanosomosis [143]. Extramedullary erythropoiesis has also been studied using an scRNA-seq approach in the experimental *T. (b.) evansi* infection model in mice. This study allowed to track the infection-associate fate of hematopoietic stem and progenitor cells (HSPCs), pro-erythroblast (Pro E), and basophilic erythroblast (Baso E). This confirmed that *T. (b.) evansi* infection blocks the production of fully mature red blood cells in the spleen. In turn, this leads to a lack of circulating red blood cells and a consequent anemia that limits oxygen supply to tissues. Under such conditions, a catabolic muscle disorder induced by lactic acidosis sets in, which is characteristic of the wasting syndrome seen in AT in trypanosusceptible livestock [132].

### 6.3. Bone Marrow

The bone marrow is the primary site of hematopoiesis, the process of generating blood cellular components, and contains hematopoietic stem cells (HSCs). HSCs mature into lymphoid lineage cells, including T, B, and NK cells, and myeloid lineage, including erythrocytes, granulocytes, and macrophages [144]. For B cells, bone marrow-derived HSCs differentiate into multipotent progenitors (MPP) and then into common lymphoid progenitors (CLP). From CLPs, pre-pro-B cells, pro-B cells, pre-B cells, and immature B cells develop. Immature B cells then migrate from the bone marrow to the spleen as transitional cells, including T1 and T2, to complete B cell development by terminal differentiation into mature B cells, i.e., MZBs or FoBs [145].

Previous cellular data has shown that *T. brucei*-infected mice suffer from a decrease in CLP progenitors as well as pre-pro-B cells, pro-B cells, pre-B cells, and immature B cells [117]. Later, it was confirmed that mice infected with *T. congolense* also suffer from a decrease in bone marrow-resident HSCs, pre-pro-B cells, pro-B cells, and pre-B cells [10], confirming the impaired B lymphopoiesis caused by trypanosome infection. As inflammation is known to promote the migration of immature bone marrow lymphocytes [146], it could be assumed that the strong type 1 inflammatory immune responses induced by trypanosomosis may contribute to pathology. Cellular data has shown that cell apoptosis, rather than migration, is the main driver of bone marrow cell loss [117]. To date, there is however, no comprehensive transcriptomic data available yet, that could confirm these cell biological observations. Due to the increased resolution of molecular techniques compared to cellular techniques such as flow cytometry, a new transcriptomic approach would likely shed more light on the ongoing development of bone marrow pathology. A first attempt in this direction was made recently by comparing transcriptomic neutrophil profiles during trypanosome infections at both spleen and bone marrow levels. Surprisingly, this data showed (i) that the 16-fold cell number increase in the spleen did not occur as a result of bone marrow-spleen cell migration and (ii) that the spleen neutrophil increase is not mirrored in the bone marrow, where only a 1.3-fold cell increase was noticed. In addition, while spleen data had shown a marked effect on neutrophil differentiation as a result of ongoing trypanosome infection, the latter had no significant effect on the overall bone marrow neutrophil differentiation pattern [119]. However, various time point assessments and additional studies in other trypanosome species are needed to obtain a more comprehensive view of how trypanosome infections affect the bone marrow cellular composition and function.

### 6.4. Adipose Tissue

One of the hallmarks of trypanosomosis is the typical weight loss syndrome, referred to as cachexia [147]. It results from the loss of muscle mass as a consequence of prolonged inflammatory anemia and the loss of adipose tissue mass. Unraveling the mechanisms that drive the latter has only recently received more attention, demonstrating that (i) it stems from the sex-dependent weight loss driven by IL-17 [148] and the shrinkage of adipocytes [30] and (ii) that adipose tissue is a significant reservoir for African trypanosomes, with parasite numbers being up to 60-fold higher than those found in solid organs or tissues such as the brain, heart, lung, or kidney [26,149]. Adipose tissue comprises various cell populations, including adipocytes, precursors, endothelial cells, mesothelial cells, smooth muscle cells, and immune cells, playing a pivotal role in numerous biological pathways [150,151,152,153]. It is a metabolically dynamic energy depot and an immunological organ [153], categorized into two predominant types: white adipose tissue and brown adipose tissue [154]. White adipose tissue constitutes a significant portion of mammalian adipose tissue, functioning as an energy reservoir through lipogenesis and lipolysis. Conversely, brown adipose tissue contributes to non-shivering thermogenesis [155]. Additionally, beige adipocytes in subcutaneous adipose tissue possess characteristics of both white and brown adipocytes, playing a role in thermogenesis within white adipose tissue [156]. Given the propensity of African trypanosomes to infiltrate white adipose tissue, scRNA-seq technology has been adopted to study the unique nature of the parasites found here and the effect of infection on specific local responses. This has demonstrated the existence of a distinct trypanosome population referred to as adipose tissue forms (ATFs) that, despite sharing the slender form morphology with bloodstream forms (BSFs), exhibit a 20% different gene expression pattern [26]. ATFs infiltrate, reside within adipose tissue, and adapt to this tissue environment by regulating the expression of genes related to cell cycle and cell signaling. Specifically, the upregulation of numerous metabolic pathways encompasses the pentose-phosphate, purine salvage, and lipid and sterol metabolic pathways. It results in the ability to utilize fatty acids as a carbon source in ATFs [157].

Integrated spatial transcriptomic results and bulk RNA-seq data also reveal that trypanosome infection alters adipose cell distribution and migration of resident and foreign cells into adipose tissue [67,149,158]. This coincides with the upregulation of inflammation-related genes, as well as the downregulation of genes primarily linked to metabolite biosynthesis and other metabolic changes. In the myeloid compartment, the analysis revealed an increase in monocytes, macrophages, and CD207^+^ Langerhans Cells. Interstitial preadipocyte 2 (IPA2) expression was identified as a hallmark of the changes that occurred in the stromal cell compartment, with the latter being a primary activator of cytokines and the antigen presentation process [67]. Infection also induces an accumulation and enrichment of T cells in adipose tissue [67,149,158]. Interestingly, transcriptomic analyses revealed specific enrichment of T cell-related transcripts in infected male mice, in inguinal white adipose tissue, not observed in females [158]. Here, an increase in CD8^+^ T cells was observed immediately following infection, while CD4^+^ T cells only began to increase significantly in the later stages. Combined results suggest that in adipose tissue, CD4^+^ T cells tend to differentiate into Th1 cells, which may substantially contribute to local IFN-γ production [149]. Additionally, flow cytometry results align with an increase in effector T cells, corresponding to an increase of TNF and IFN-γ in gonadal adipose tissue at the transcriptomic level, corroborating the induction of a Th1-type response [149,158]. Furthermore, Treg-associated results align with increased expression of FOXP3, CTLA-4, and IL-10 in gonadal adipose tissue, suggesting a contribution of these cells in controlling excessive inflammation and supporting the parasite’s survival [149]. Bulk RNA data analysis also reveals a strong expression of genes related to Th17 differentiation in inguinal white adipose tissue, an observation confirmed by flow cytometer data. Interestingly, a previous study indicated an increase in the number of interstitial preadipocytes, accompanied by an increase in the expression of *Il17ra* within these preadipocytes. These results suggested that IL-17A/F signaling, controlled by T cells and adipocytes, helps to control the local parasite burden, and prolongs host survival. Additionally, *Il17af^−/−^* mice (IL-17A and IL-17F deficient mice) and *Adipoq^Cre^* x *Il17ra^fl/fl^* mice (*Il17ra*-deficient in white adipocytes mice) exhibited an increased body weight and food intake after trypanosome infection. These results investigate the vital role of IL-17 in controlling adipose tissue dynamic and parasite burden during trypanosomosis [158].

### 6.5. Brain

Upon entering the host through the bite of the tsetse fly, African trypanosomes navigate a complex journey through the body [159]. They traverse the skin, travel via the bloodstream, and sequentially infiltrate diverse organs and tissues, including adipose tissue, kidney, lungs, and spleen, finally reaching the brain during the chronic stage [26,119,160,161]. This infiltration into the brain leads to consequential outcomes, including sleep disturbances and decreased motor function. If left untreated, these infections will mostly be fatal [162,163]. Given the intricacies of the brain, comprising diverse cell populations such as oligodendrocytes, astrocytes, microglia, and ependymal cells, and its significance as a vital organ with various functional regions (including the limbic cortex, frontal cortex, paleocortex, temporal cortex, parietal cortex, basal forebrain, thalamus, hippocampus, hypothalamus, cerebellum, midbrain, pons, and spinal cord), studying the crosstalk between the brain and parasites during infection becomes a challenging yet essential endeavor [164,165]. To tackle this challenge, combined single-cell and spatial transcriptomic analysis has enabled the in-depth exploration of gene expressions within the diverse cell type population, the heterogeneity of each cell type in response to pathogen invasion, and the precise location of each gene population and cell type within the intricate structure of the brain. This recently has yielded a comprehensive understanding of the localization and adaptation of trypanosomes within the brain, showing that slender and stumpy form parasites display distinctive distributions in the brain [82]. Distinct slender and stumpy forms of *T. brucei* exhibit specific spatial distributions in the brain, with slender forms favoring regions such as the cerebral caudoputamen, corpus striatum, thalamus, hippocampus, cerebral cortex, hypothalamus, and circumventricular organs (CVOs) [82,166]. Staining with smFISH confirms an increased parasite presence near CVOs. Additionally, gene ontology and pathway analyses reveal an overrepresentation of genes related to translation, gene expression control, and biosynthetic processes, indicating a targeted strategy upon breaching the blood–brain barrier [82]. The hypothalamic immune response to *T. brucei* infection involves myeloid (microglia, monocytes, macrophages), T cells (follicular-like regulatory CD4^+^ T cells, and cytotoxic CD8^+^ T cells), and B cells (CD138^+^ plasma cells) [82]. Myeloid subsets near CVOs display upregulated transcriptional programs linked to antigen presentation and adaptive immune response development. Cytotoxic CD8^+^ T cells are exclusively found in the brain parenchyma, while regulatory CD4^+^ T cells prevail in specific regions [82]. Cell–cell interaction analyses reveal T cell recruitment and activation in the brain during chronic infection, mediated by stromal cell signaling [82]. Interestingly, CD138^+^ plasma cells in the B cell compartment show increased surface markers and transcription factors during infection, confirmed by flow cytometry in chronic infections. Ligand-receptor interaction analysis highlights the upregulation of the pro-survival factor *Tnfsf13b* by homeostatic microglia upon infection, interacting with its receptor, *Tnfrsf17*, highly expressed in CD138^+^ plasma cells [82]. Combined, these results unveil intricate cellular responses and spatial dynamics in the brain during *T. brucei* infection, emphasizing the reciprocal interaction between microglia and CD138^+^ plasma cells.

Trypanosomes accumulate in the CNS and meningeal spaces, causing meningitis [167,168]. Using combining single-cell transcriptomics and mass cytometry by time-of-flight (CyTOF), it was found that chronic infection with *T. brucei* causes a broad rearrangement of the immune landscape and the development of ectopic lymphoid aggregates (ELAs) in murine meninges [116]. The formation of ELAs, which consists of ER-TR7^+^ fibroblastic reticular cells, CD21/35^+^ follicular dendritic cells, CXCR5^+^ PD1^+^ T follicular helper-like phenotype, GL7^+^ CD95^+^ GC-like B cells, and plasmablasts/plasma cells, has been reported in autoimmune diseases, including neuropsychiatric lupus [169] and multiple sclerosis [170]. Furthermore, B cells in the infected meninges produced high-affinity autoantibodies that could recognize mouse brain antigens in an LTβ signaling-dependent manner, consistent with the finding of the presence of autoreactive IgG antibodies in the CSF of second-stage HAT patients [116]. Several host factors, including myelin basic protein recognized by these autoantibodies, contribute to cortical demyelination and brain pathology [114,115]. These results show chronic meningitis caused by trypanosomosis, resulting in the development of impaired peripheral tolerance and autoimmunity.

### 6.6. Lung

The lungs are the vital mammalian respiratory organs sustaining aerobic life. They also constitute a unique immunological organ with a huge mucosal surface contact zone with the outside world, 200 times larger than the skin. The lung contains a varied community of tissue-resident innate and adaptive immune cells, cooperating to preserve tissue equilibrium and defend against recurring pathogen challenges. The immune cell architecture of the lungs comprises an extensive network of innate immune cells and an adaptive immune response that includes alveolar macrophages, dendritic cells, neutrophils, B cells, T cells, and antigen-presenting cells [171,172]. Specific populations of resident immune cells sustain immune homeostasis in various lung compartments. The alveolar macrophages and dendritic cells residing in the lungs, part of the swift innate immune system, engage in phagocytosis of antigens. Alveolar macrophages focus on clearance, while dendritic cells process and present antigens to T cells in the lymph nodes, draining the lungs. In the initial defense against antigenic particulates, innate cells, innate-like γδ T cells, and innate lymphoid cells form a swift line of defense. T cells, activated in response to specific antigens, travel through the lymphatics and pulmonary capillaries, reaching the lung parenchyma and infection site. Additionally, B cells contribute to the immune response by generating antibodies through the differentiation and activation of plasma B cells [173]. Maintaining a balance in immune responses is essential for immune homeostasis in the lungs.

Lung-residing trypanosomes constitute approximately one-fifth of the total tissue burden at the onset of infection and during disease progression [26,71,174]. The relative importance of pulmonary trypanosome presence was underscored by a notable increase in the parasite burden as infection progressed. This suggests that the lungs are a favorite organ for the multiplication and survival of these parasites. *T. b. brucei* can rapidly and permanently colonize the lungs. They occupy the extravascular space surrounding blood vessels of the alveoli and bronchi. These parasites form nests in these lung areas and actively multiply. Their close association with collagen makes an extracellular matrix structure, suggesting an adaptation to this specific tissue environment. This phenomenon of activated replication within the lung tissue likely contributes to the increased parasite burden in the observed infection process. Interestingly, there is no apparent infection-associated pulmonary dysfunction in mice in the context of *T. b. brucei* infection. However, trypanosomosis could have significant consequences for the host immune system and the host’s ability to mount an effective immune response against opportunistic pathogens [174].

To better understand the lung immunology response to trypanosomes, a Nanostring nCounter digital transcriptomic approach has been used to analyze RNA extracted from lung tissues of *T. b. brucei* infected mice, revealing a significant upregulation of negative immune checkpoint regulators, implying an inhibition or dampening of T cell functions. Interestingly, *Foxp3* was mainly expressed in a subset of CD4^+^ T cells but not differently regulated in the lung during infection, suggesting a potential uncoupling of traditional regulatory T cell responses. The gene ontology analysis confirms major changes in negative regulation of T cell activation, indicating potential modulation of T cell responses. In addition, a strong induction of γδ T cell responses suggests an active role of these cells in the immune reaction [174]. The specific mechanisms by which γδ T cells contribute to disease control in trypanotolerant animals may involve their ability to recognize and respond to trypanosome antigens, as described previously [67]. B cell memory depletion and alterations in the B cell receptor signaling pathway also suggest a change in the B cell’s responses during infection. In the context of the lung neutrophils chemotaxis, migration was not observed during the early stage, while only a marginal cell number increase occurred in the later stage of infection. In this study, a strong classic macrophage polarization was observed. Hierarchical clustering analysis revealed the induction of an inflammatory response involving IFN-γ, TNF-α, IL-2, and IL-6. In addition, other genes that were affected by infection were *Cxcl13*, *Ms4a7*, *Pdcd1lg2*, *Nos2*, and *ccl8*, all signature markers for inflammation, cell recruitment, and cell migration, in particular, with respect to monocyte/macrophage mobility. Interestingly, some key markers genes were also identified based on their downregulation pattern, including *Cfd*, *Ackr4*, *Cxcl5*, and *Hamp*, involved in T cell migration, neutrophil migration, and the regulation of iron loading of macrophages and inflammatory events related to anemia and hypoxia. This comprehensive analysis revealed a complex immune response to *T. b. brucei* infection in the lung, characterized by a strong pro-inflammatory environment, alterations in T cell regulation, especially significant involvement of γδ T cells, B cell responses depletion, and classic macrophage activation [174].

### 6.7. Liver

The liver is a highly metabolically active organ with many functions, including regulating lipid and carbohydrate metabolisms, metabolic detoxification, protein synthesis, and bile secretion. Hence, the liver assumes a crucial role in the storage, synthesis, and release of lipids, helping to improve overall lipid balance in the body [175,176,177]. The liver’s architecture encompasses various resident cell types that play a crucial role in innate immunity, including hepatocytes, Kupffer cells, NK cells, NK-T cells, hepatics dendritic cells, stellate cells, and liver sinusoidal endothelial cells [178]. Approximately 80% of the blood supply to the liver arrives through the hepatic portal vein, carrying blood that is low in oxygen but rich in nutrients and molecules from the intestinal microbiota or parasites. The liver’s unique anatomical characteristic exposes this organ to high levels of pathogenic components that would be recognized as danger signals and potent pro-inflammatory stimuli in peripheral tissue. Consequently, the liver must effectively distinguish pathogenic damage-associated molecular patterns, especially pathogen-associated molecular patterns. Hence, despite being a non-lymphoid organ, the liver can play a pivotal role in the immune response against pathogens [179]. This has been well documented at a cellular level for experimental trypanosome infections, where the liver was shown to be crucial for periodic peak parasitemia removal and pathology regulation [70,180,181]. The immune response to trypanosome infection in the liver primarily involves Kupffer cells [182], constituting approximately 80–90% of the total tissue macrophage in the body [183]. Despite the liver’s crucial role, virtually no in-depth transcriptomic data is available to allow to validate previous work based on cell morphological work. Only few studies are available, using cDNA microarray to assess the gene expression profile during early stage infection with *T. b. brucei* [181] and *T. (b.) evansi* [180]. The obtained data suggests that *T. (b.) evansi* has a more dominant impact on the liver than *T. b. brucei* but taking that only one stain of each parasite was studied, the general validity of this broad conclusion could be called into question. Due to *T. brucei* infection, genes were upregulated in the liver, including *Hsp70*, *chemokine ligand 9*, *chemokine ligand 7*, *B-cell leukemia/lymphoma*, *Bcl2-associated X protein*, *caspase 3*, and *zinc finger protein 410*. Proteins encoded by these genes are primarily involved in the immune and inflammatory responses as transcription factors or nucleic acid-binding proteins. In contrast, downregulated genes included those coding for the IL-1 receptor accessory protein, proliferating cell nuclear antigen (Pcna), cytochrome P450, solute carrier family 6, acid phosphatase 2, and alkaline phosphatase 2. These genes mainly encode factors involved in protein amino acids phosphorylation and dephosphorylation, transduction signal, metabolism regulation, and protein biosynthesis. As for Hsp70, a molecular chaperone that assists in folding newly synthesized proteins and refolding misfolded or denatured proteins, the interaction of this protein with co-chaperones that has been considered a potential drug target for *T. b. brucei* is noteworthy [181].

## 7. Conclusions

To date, the wider implementation of scRNA-seq and associated technologies has only found a limited entrance into trypanosome immunobiology research, but the studies carried out so far have already provided new insights into events such as (i) infection associated brain inflammation, (ii) B cell dysfunction, (iii) T cell regulation and (iv) macrophage activation (Table 1). Taken that there is still no proven vaccine strategy available that can help control trypanosomosis under field conditions, continued progress in detailed transcriptomic research might help break this deadlock in the future. There are, however, multiple issues to be considered here. 

First, trypanosomosis is a systemic infection, affecting every single organ of the body, as well as every immune compartment. That means that while a recent focus on brain, spleen, lung, and adipose tissue infections has provided new insights, more work remains to be completed with respect to the liver, kidneys, testes, ovaries, gut, lymph nodes, and all the other niches where trypanosomes might have a particular effect. Combining all this data to obtain a more holistic view of the true biological effect of trypanosomosis on the mammalian body will be a next-level challenge. 

Second, most current studies are carried out with a single trypanosome isolate, mainly focusing on *T. b. brucei* and, to a lesser extent, on *T. (b.) evansi* and *T. congolense*. However, it is clear that in order to make a statement with a true general level of validity, multiple isolates of the same parasite need to be compared. Few studies have focused on the immunology of HAT due to limited human data. In particular, human data on *T. b. rhodesiense* HAT with high virulence characteristics are more limited compared to *T. b. gambiense* HAT. Therefore, studies on *T. b. rhodesiense* strains with varying virulence from different *T. b. rhodesiense* isolates in experimental animal models will be valuable in investigating a range of different infection outcomes in HAT. 

Third, both animal trypanosomosis and human *T. b. gambiense* HAT are chronic infections. That means that in order to obtain a better-detailed view of disease progression, chronic infection models need to be studied, in which multiple time points can be compared. Ideally, such experimental data should then be compared to biopsy material of infected animals or human samples, although the latter poses multiple challenges. 

Fourth, the diversification of infection routes in experimental AT is required. Although there is research on AT undertaken in cattle showing that a *T. congolense* cysteine protease (congopain) elicits a high IgG response in trypanotolerant N’Dama cattle, unlike trypanosusceptible Boran cattle [184], most studies on AT have been conducted in experimental animal models, especially mice for logistical reasons. In particular, most experimental studies on trypanosomes have directly inoculated bloodstream forms of parasites into mice via the intraperitoneal or intravenous routes and bypassed the skin. However, natural infection with AT begins when parasites are injected intradermally into a mammalian host by an infected tsetse fly. Previous studies showed that inoculation routes can affect the outcomes of infection [185,186]. Mice infected by intradermal inoculation were less susceptible to trypanosome infection than mice infected by intraperitoneal inoculation thanks to the potential local immune response in the skin [185]. Furthermore, tsetse fly saliva is known to promote the blood-feeding process, facilitate parasite transmission, and modulate the immune response of the host [187,188]. Therefore, in addition to broadening the scope of research, such as the study of parasite-host interactions in livestock, using various infection routes, including the intradermal route, will enable the capture of early events that occur in natural infection with AT in animals, including livestock.

Finally, it has been known for years that various trypanotolerant animal breeds exist and that also, in the case of HAT, some individuals exhibit very prolonged asymptomatic infections, sometimes lasting “forever”. It would be interesting if comparisons between susceptible and tolerant “conditions” were addressed in future approaches using high throughput transcriptomics, eventually combined with metabolomics, proteomics, and lipidomics. Knowledge of secondary metabolites, which may be products of the primary mechanism for host–pathogen adaptation [189], can improve our understanding of the chemical mechanisms of trypanosome-relevant host interactions. *T. congolense* and *T. vivax* are the main livestock trypanosomes. In particular, since *T. vivax* is virtually unable to grow in mice, it remains a less-studied *Trypanosoma* species despite being an important livestock trypanosome. However, phenolic compounds were found in the breath and urine of cows infected with *T. congolense* and *T. vivax*, and using these compounds, trypanosomosis was identified with high sensitivity and specificity under low-level infections that are not detectable by microscopy [190]. In addition, stage determination in HAT is a key element of disease management, and the determination currently requires microscopic evaluation of CSF after lumbar puncture. Nevertheless, metabolites that could be used as biomarkers to distinguish the presence of disease and disease stage were discovered in plasma through a metabolomics approach, and it offers the possibility of eliminating the need for diagnostic lumbar punctures in the future [191]. Therefore, metabolites from trypanosome-infected humans and animals as biomarkers enable the detection of asymptomatic or latent infections and can be used for rapid, reliable, and non-invasive trypanosomosis diagnosis under field conditions.

Meanwhile, disease states, including infectious diseases, are accompanied by changes in gene expression and/or their protein products. Human and animal physiology is a highly complex system coordinated by the response and adaptation to various internal and external stimuli, and single techniques alone are insufficient to comprehensively understand the physiology and pathophysiology. Therefore, access to various ‘omics’ platforms will enhance the comprehensive understanding of AT, including HAT. However, it should be noted that although transcriptomics is a robust, high-throughput, cost-effective technology, the qualified tens of thousands of mRNAs may not necessarily be sufficient to predict protein abundance accurately [192,193]. Post-transcriptional events, such as alternative mRNA splicing, increase the diversity of proteins from a fixed number of genes, and protein turnover, specific proteolytic processing, and post-translational modifications also affect the level of protein expression [194]. In humans, approximately 50% of genes are believed to undergo alternative splicing [195]. It is difficult to accurately predict protein profiles from transcriptomic results if these events are not taken into account, so it is important to consider transcriptomic and proteomic data together [196]. Therefore, integrated analysis of transcriptomic and proteomic datasets that improves understanding of the relationship between transcriptomes and proteomes can be an approach to improve biological insights [197,198]. While still utopic at this moment, such an approach would be ideal to curb the threat trypanosomosis poses to livestock agriculture and human health in general.

## Figures and Tables

**Figure 1 pathogens-13-00188-f001:**
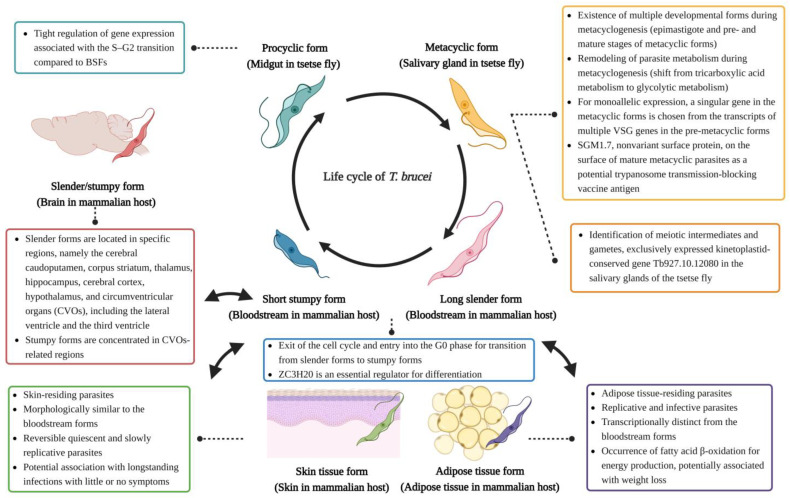
Simplified life cycle of *T. brucei* and particular forms residing in host tissue. *Trypanosoma brucei* is a parasite that uses the tsetse fly as its definite host, colonizing the midgut and salivary glands. Infection of the mammalian ‘reservoir’ is accomplished by injection of metacyclic parasites that quickly transform into dividing long slender bloodstream forms (BSFs). From here, tissues are invaded, such as the skin and adipose tissue, where transcriptome research has allowed to identify uniquely adapted parasite forms such as skin tissue forms (STFs) and adipose tissue forms (ATFs). Ultimately, the mammalian brain is invaded where the main pathology that characterizes HAT occurs (figure created using BioRender.com).

**Figure 2 pathogens-13-00188-f002:**
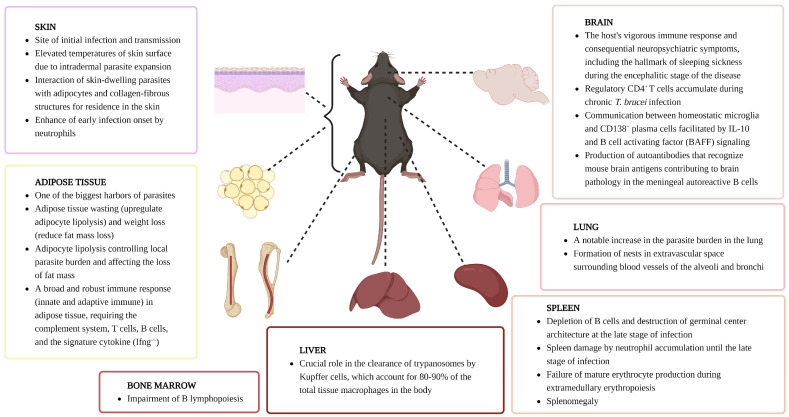
Overview of trypanosome–host interactions in different organs. Trypanosomes colonize multiple organs of the mammalian hosts and invade tissues through blood and lymphatic vessels. This results in local modulations of host immunity and subsequent pathological outcomes (figure created using BioRender.com).

**Table 1 pathogens-13-00188-t001:** Overview of transcriptomics studies on trypanosomosis.

Trypanosome Species	Host	Organ	Targeted Cells	Outcome of Infection	Technology	References
*Trypanosoma b. rhodesiense*	Human	Blood/Cerebral spinal fluid	Not available	Upregulation of gene expression related to the innate immune response in the early stage of infection. Upregulation of gene expression related to the anti-inflammation and neuro-degeneration in the late stage of infection.	RNA-seq (Illumina NextSeq500 System)	[56]
*Trypanosoma (b.) evansi*	Mouse	Spleen	B cells (transitional B cells, B1 cells, atypical memory B cells, marginal zone B cells, follicular B cells, germinal center-like B cells, plasma cells)	Rapid activation of mature B cells (GC-like B cells, atypical memory B cells, and plasma B cells). Depletion of the entire B cell compartment. Improved parasitemia control and maintenance of the B cell compartment in the infected AID^−/−^ mice.	scRNA-seq (10× Genomic Chromium Controller/ Illumina NovaSeq6000 System)	[107]
*Trypanosoma (b.) evansi*	Mouse	Spleen	CD4^+^ T cells (naïve CD4^+^ T cells, Th1 cells, regulatory T cells, T follicular helper cells, cycling T cells, myeloid cell phenotype expressing T cells)	Th1 cells simultaneously produce IFN-γ and IL-10. Prediction of the role of T follicular helper cells in abrogation of germinal center formation.	scRNA-seq (10× Genomic Chromium Controller/ Illumina NovaSeq6000 System)	[96]
*Trypanosoma (b.) evansi*	Mouse	Spleen	Macrophages (monocytes- derived macrophages, erythroblast island macrophages, tissue resident macrophages), Erythroid cells (early and late erythroid precursor cells, pro-erythroblast, basophilic erythroblast, polychromatophilic erythroblast, orthochromatic erythroblast)	Tissue-resident macrophages simultaneously express markers of both classic and alternatively activated cells. Inhibition of the production of fully mature red blood cells during extramedullary erythropoiesis.	scRNA-seq (10× Genomic Chromium Controller/ Illumina NovaSeq6000 System)	[132]
*Trypanosoma (b.) evansi*	Mouse	Bone marrow	B cells (early B lineage cells)	Loss of B cell replenishment.	scRNA-seq (10× Genomic Chromium Controller/ Illumina NovaSeq 6000 System)	[107]
*Trypanosome b. brucei*	Mouse	Bone marrow	Neutrophils (pre-neutrophils, immature neutrophils, mature neutrophils)	No significant effect on the overall pattern of neutrophil differentiation.	scRNA-seq (10× Genomic Chromium Controller/ Illumina NovaSeq 6000 System)	[119]
*Trypanosoma b. brucei*	Mouse	Spleen	Neutrophils (pre-neutrophils, immature neutrophils, mature neutrophils), Plasma cells	Expansion of splenic neutrophil subpopulations and classification in four previously defined subpopulations. Prevention of organ damage, increase in plasma cells and prolonged host survival after neutrophil depletion.	scRNA-seq (10× Genomic Chromium Controller/ Illumina NovaSeq 6000 System)	[119]
*Trypanosome b. brucei*	Mouse	Adipose Tissue	Neutrophils, Macrophages, Monocytes, CD4^+^ T cells, CD8^+^ T cells, B cells	Lymphocytes produce type 1 effector cytokines, namely IFN-γ and TNF-α. The accumulation of antigen-specific IgM and IgG antibodies in adipose tissue. Inducing a broad and robust immune response in adipose tissue, requiring the complement system to reduce parasite burden locally, and emphasizing the critical role of T and/or B cells and the signature cytokine (Ifng^−/−^) in the adaptive immune system’s response.	RNA-Seq (Illumina HiSeq 2000 System)	[149]
*Trypanosome b. brucei*	Mouse	Adipose Tissue	Adipocyte, T cells (Th17 cells, CD27^−^ Vɣ6^+^ cells)	IL-17, derived from Th17 and Vγ6^+^ cells, orchestrates the immune response, culminating in adipose tissue wasting and weight loss.	RNA-Seq (Illumina NovaSeq 6000 System), scRNA-seq (10× Genomic Chromium Controller/ Illumina NovaSeq 6000 System)	[158]
*Trypanosome b. brucei*	Mouse	Brain	Myeloid (homeostatic microglia, monocytes, border-associated macrophages, infection-associated mononuclear phagocytes), T cells (follicular-like regulatory CD4^+^ T cells, cytotoxic CD8^+^ T cells), B cells (CD138^+^ plasma cells)	Glial responses to infection occur near circumventricular organs, including the lateral and third ventricles. Crosstalk between homeostatic microglia and CD138^+^ plasma cells, facilitated by IL-10 and B cell activating factor (BAFF) signaling.	scRAN-seq with spatial transcriptomics (10× Genomic Chromium Controller/ Illumina NextSeq 550 System)	[82]
*Trypanosome b. brucei*	Mouse	Brain	Meningeal fibroblasts, CD21/35^+^ follicular dendritic cells, CXCR5^+^ PD1^+^ T follicular helper-like phenotype, GL7^+^ CD95^+^ GC-like B cells, plasmablasts/plasma cells	Immune landscape remodeling in the meninges containing T follicular helper cells-like T cells, GL7^+^ CD95^+^ GC-like B cells, and plasmablasts/plasma cells. Accumulation of meningeal autoreactive B cells that generate autoantibodies recognizing a broad range of host antigens, including myelin basic protein.	scRNA-seq (10× Genomic Chromium Controller/ Illumina NovaSeq 6000 System)	[116]
*Trypanosome b. brucei*	Mouse	Lung	T cells (γδ T cells), NK cells, Macrophages, B cells	The presence of trypanosomes is non-detrimental to lung function. Co-infections could affect susceptibility for opportunistic infections. Negative regulation of T cell activation, strong induction of γδ T cell. B cell memory depletion. Strong macrophage polarization towards classic activation.	NanoString nCounter (nCounter MAX Analysis System)	[174]
*Trypanosoma b. brucei*	Mouse	Skin	IL-17A-producing Vγ6^+^ cells	Expansion of dermal IL-17A-producing Vγ6^+^ cells (γδ T cell subsets) in subcutaneous adipose tissue. Potential interactions between subcutaneous adipocytes and Vγ6^+^ cells.	scRAN-seq with spatial transcriptomics (10× Genomic Chromium Controller/ Illumina NovaSeq6000 System)	[67]

Manufacturer of applied technology: Illumina HiSeq 2000 System, Illumina, CA, USA; Illumina NextSeq500 System, Illumina, CA, USA; Illumina NovaSeq6000 System, Illumina, CA, USA; nCounter MAX Analysis System, NanoString Technologies, WA, USA; 10× Genomic Chromium Controller, 10× Genomics, CA, USA.

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
