# Peer review of "Advances in the Immunology of the Host–Parasite Interactions in African Trypanosomosis, including Single-Cell Transcriptomics"

_pathogens, 2024, doi:10.3390/pathogens13030188_

Round 1

Reviewer 1 Report

Comments and Suggestions for Authors

I want to start by saying that I think this is a really excellent review of the field of single cell transcriptomics in relation to trypanosomiasis. It covers really key topics, whilst also linking to the seminal historical work that ultimately led to recent use of single cell to explore these questions in more detail. I particularly liked that at the end of the review, it lays out the areas where our knowledge is still lacking, such as the liver and kidneys, and hopefully this will prompt research teams to begin investigating these tissues in more detail.

Line 123-125: This is a fascinating idea, that mammals are actually the vector, and the goal is to propagate and increase genetic diversity within the fly.

Line 144-149: Indeed, Trindade et al did show that ATFs upregulate genes associated with fatty acid oxidation, but in a later paper from the lab, by Machado et al, they propose that fatty acids are actually toxic for T. brucei and that lipolysis protects mice against trypanosomiasis. It would be useful to mention this here, and discuss it briefly.

Lines 296-307 are absent of any citations. Please include citations in this section.

Line 323-327: This sentence requires a citation. Also in lines 326-327, the sentence is incomplete.

Line 337-341: Please include a citation.

Line 389: In the section on B cells starting on this line, there is no mention of the recent work of Quintana et al on autoreactive B cells in the meninges during T. brucei infection. Furthermore, this work should be discussed in the section on the brain (line 759 onwards) and referred to in Table 1.  

Line 410-412: Please include a citation

Line 702-706: The shrinkage of adipocytes during T. brucei was first characterised by Sinton et al in their preprint (September 2022), prior to Machado et al in their preprint (November 2022), and so the references should be updated to reflect this.

Line 751-752: Just a small thing here. In this line, it should state that there is an increase in the number of interstitial preadipocytes and that they increase expression of Il17ra. IL-17ra should be changed to Il17ra

Line 778: This line isn’t cited correctly as it refers the reader to citation 158. The preceding sentence is referring to spatial and single cell sequencing, and how this has improved understanding of the distribution and localisation of trypanosomes in the brain. However, citation 158 doesn’t utilise these technologies so it’s not appropriate here.

Line 778-818: This section is lacking in citations, and it would be really helpful to include them. Moreover, recent work by Quintana et al (2023) used single cell sequencing to examine the meninges during T. brucei infection. It would be useful to include a discussion of this work in this section too.

Table 1: It would be helpful to include details of Quintana et al (2023) PLOS Biology on single cell of the meninges.

Author Response

We are very grateful for the reviewers' thoughtful and helpful comments, which have helped improve the quality of our paper.

Reviewer 1:

I want to start by saying that I think this is a really excellent review of the field of single cell transcriptomics in relation to trypanosomiasis. It covers really key topics, whilst also linking to the seminal historical work that ultimately led to recent use of single cell to explore these questions in more detail. I particularly liked that at the end of the review, it lays out the areas where our knowledge is still lacking, such as the liver and kidneys, and hopefully this will prompt research teams to begin investigating these tissues in more detail.

We would like to express our gratitude to the reviewer for the overall positive evaluation of the quality of our paper. We would also like to thank the reviewers for reading the paper carefully and suggesting in detail what we need to improve, including referencing most recent papers.

Line 123-125: This is a fascinating idea, that mammals are actually the vector, and the goal is to propagate and increase genetic diversity within the fly.

Once again, we thank the reviewer for the positive feedback.

Line 144-149: Indeed, Trindade et al did show that ATFs upregulate genes associated with fatty acid oxidation, but in a later paper from the lab, by Machado et al, they propose that fatty acids are actually toxic for T. brucei and that lipolysis protects mice against trypanosomiasis. It would be useful to mention this here, and discuss it briefly.

(Please note that the line numbers have changed after making requested changes to the document) We added new content (Machado et al) on adipocyte lipolysis (citation 30) in lines 149-153.  The content of this paper was also used to updated in Figure 2 (line 483).

Lines 296-307 are absent of any citations. Please include citations in this section.

We included a number of new citations (citations 64-72) throughout this paragraph in lines 302-312.

Line 323-327: This sentence requires a citation. Also in lines 326-327, the sentence is incomplete.

We included an appropriate citation (citation 75) and double-checked the sentences in lines 328-332.

Line 337-341: Please include a citation.

We added a citation (citation 82) to support the sentence in line 346.

Line 389: In the section on B cells starting on this line, there is no mention of the recent work of Quintana et al on autoreactive B cells in the meninges during T. brucei infection. Furthermore, this work should be discussed in the section on the brain (line 759 onwards) and referred to in Table 1. 

We would like to thank the reviewer for introducing a new, up-to-date paper. We added information about this paper to the B cells section (5-3. B cells) in lines 443-447 alongside the brain section (6-5. Brain) and Table 1 and additionally cited the same paper after the existing sentence in line 462 (citation 116). The content of this paper was used to updated in Figure 2 (line 483).

Line 410-412: Please include a citation

We added a citation (citation 102) to support the sentence in line 418.

Line 702-706: The shrinkage of adipocytes during T. brucei was first characterised by Sinton et al in their preprint (September 2022), prior to Machado et al in their preprint (November 2022), and so the references should be updated to reflect this.

Line 751-752: Just a small thing here. In this line, it should state that there is an increase in the number of interstitial preadipocytes and that they increase expression of Il17ra. IL-17ra should be changed to Il17ra

We thank the reviewer for the thorough correction. We corrected the sentence in lines 763-764.

Line 778: This line isn’t cited correctly as it refers the reader to citation 158. The preceding sentence is referring to spatial and single cell sequencing, and how this has improved understanding of the distribution and localisation of trypanosomes in the brain. However, citation 158 doesn’t utilise these technologies so it’s not appropriate here.

We appreciate for the reviewer’s attentiveness. We deleted citation 158 from that sentence in line 790.

Line 778-818: This section is lacking in citations, and it would be really helpful to include them. Moreover, recent work by Quintana et al (2023) used single cell sequencing to examine the meninges during T. brucei infection. It would be useful to include a discussion of this work in this section too.

We added new citation in line 772 (citation 159) and significantly modified the brain section. The previously lengthy explanation of the paper by Quintana et al (citation 82) has been written more concisely in lines 790-812. In addition, we described the new paper (citation 116) the reviewer suggested in the B cells section (5-3. B cells) and in more detail in the brain section (6-5. Brain) in the lines 813-828.

Table 1: It would be helpful to include details of Quintana et al (2023) PLOS Biology on single cell of the meninges.

We included the contents of this citation (citation 116) in Table 1 as well.

Reviewer 2 Report

Comments and Suggestions for Authors

Literature reviews by Choi et al., are highly useful assets for the scientific community. Due to the rapid acceleration of research in recent years and the increasing number of original papers being published, review articles have gained significance as a means to stay up to date about advancements in a specific research field. A good review article provides readers with an in-depth understanding of a field and highlights key gaps and challenges to address with future research, which this review tried but some more gap and proposed solutions should be included. For example we should include when parasite host interaction the metabolomics aspect should be clearly stated as genes do not interact directly with the environment but using metabolites which are effectors need to be discussed, the field is at its early stage but there is some indication for instance  https://doi.org/10.3389/fmicb.2022.922760.   

The other thing is the need for relevant parasites (actively circulating in the host) and livestock host interaction should be emphasized for future sequencing studies as information we get from mice is not translatable to livestock for instance.

Comments on the Quality of English Language

NA

Author Response

We are very grateful for the reviewers' thoughtful and helpful comments, which have helped improve the quality of our paper.

Reviewer 2:

Literature reviews by Choi et al., are highly useful assets for the scientific community. Due to the rapid acceleration of research in recent years and the increasing number of original papers being published, review articles have gained significance as a means to stay up to date about advancements in a specific research field. A good review article provides readers with an in-depth understanding of a field and highlights key gaps and challenges to address with future research, which this review tried but some more gap and proposed solutions should be included. For example we should include when parasite host interaction the metabolomics aspect should be clearly stated as genes do not interact directly with the environment but using metabolites which are effectors need to be discussed, the field is at its early stage but there is some indication for instance  https://doi.org/10.3389/fmicb.2022.922760.  

We would like to express our gratitude to the reviewer for the kindly positive feedback on our paper and for suggesting direction for writing a good paper. In particular, by referring to the paper the reviewer provided as an example (https://doi.org/10.3389/fmicb.2022.922760), we were able to rethink about the importance of research in the context of metabolomics. Therefore, in order to emphasize the importance of not only transcriptomics but also metabolomics, another 'omics' technology, we added the paper the reviewer gave as an example (citation 190) and a paper that studied metabolites in HAT patients (citation 191) in lines 983-999.

The other thing is the need for relevant parasites (actively circulating in the host) and livestock host interaction should be emphasized for future sequencing studies as information we get from mice is not translatable to livestock for instance.

For the study of African trypanosomosis in livestock, sequencing studies in livestock are absolutely important. However, we suggested various infection routes including the intradermal route in the experimental animal models as a realistic alternative in our paper because the intradermal route is expected to enable the study of early events occurring in natural infections. Additionally, the use of various trypanosome strains was suggested as an alternative to overcome the lack of human data for research on human African trypanosomosis. The information is described in lines 961-977 and lines 950-955, respectively.

Reviewer 3 Report

Comments and Suggestions for Authors

The review is well-written, comprehensive, and didactic, and highlights the few advances made in sc-seq transcriptomic in the organs affected by African Trypanosomes. Apart from the remarks about the limitations of the technology mentioned by the Authors in Conclusions (7), I would add that transcriptomics does not always match proteomics, nor is it always a reliable source of information on gene expression. There are abundant reports on this issue that can be googled.

I will accept this review, however: I would rephrase the work's title because it is a bit misleading. It should rest something like: “Advances in the immunology of the host-parasite interaction in African tripanosomiasis, including single-cell transcriptomics”

Revise format of References: example ref 83 

Author Response

We are very grateful for the reviewers' thoughtful and helpful comments, which have helped improve the quality of our paper.

Reviewer 3:

The review is well-written, comprehensive, and didactic, and highlights the few advances made in sc-seq transcriptomic in the organs affected by African Trypanosomes. Apart from the remarks about the limitations of the technology mentioned by the Authors in Conclusions (7), I would add that transcriptomics does not always match proteomics, nor is it always a reliable source of information on gene expression. There are abundant reports on this issue that can be googled.

We appreciate all the reviewer’s valuable comments with kindly positive feedback on our paper. Thanks to the reviewer’s deep insight, we could mention the limitations of transcriptomics as a single technology and propose integrated research with proteomics as a solution. This concept is described in lines 1000-1017.

I will accept this review, however: I would rephrase the work's title because it is a bit misleading. It should rest something like: “Advances in the immunology of the host-parasite interaction in African tripanosomiasis, including single-cell transcriptomics”

We thank the reviewer for suggesting a new title that is a very good example of what our paper is all about. We strongly agree with the reviewer's recommendation and kindly request that the title of this paper be changed to: ‘Advances in the immunology of the host-parasite interactions in African trypanosomosis, including single-cell transcriptomics’.

Revise format of References: example ref 83

We unified the styles of all references. We thank the reviewer for his careful observation.